# A Pseudo-Bayesian Algorithm for Robust PCA

**Tae-Hyun Oh**[1]    **Yasuyuki Matsushita**[2]    **In So Kweon**[1]    **David Wipf**[3]*
[1]Electrical Engineering, KAIST, Daejeon, South Korea
[2]Multimedia Engineering, Osaka University, Osaka, Japan
[3]Microsoft Research, Beijing, China
`thoh.kaist.ac.kr@gmail.com`    `yasumat@ist.osaka-u.ac.jp`
`iskweon@kaist.ac.kr`    `davidwip@microsoft.com`

## Abstract

Commonly used in many applications, robust PCA represents an algorithmic attempt to reduce the sensitivity of classical PCA to outliers. The basic idea is to learn a decomposition of some data matrix of interest into low rank and sparse components, the latter representing unwanted outliers. Although the resulting problem is typically NP-hard, convex relaxations provide a computationally-expedient alternative with theoretical support. However, in practical regimes performance guarantees break down and a variety of non-convex alternatives, including Bayesian-inspired models, have been proposed to boost estimation quality. Unfortunately though, without additional a priori knowledge none of these methods can significantly expand the critical operational range such that exact principal subspace recovery is possible. Into this mix we propose a novel pseudo-Bayesian algorithm that explicitly compensates for design weaknesses in many existing non-convex approaches leading to state-of-the-art performance with a sound analytical foundation.

## 1  Introduction

It is now well-established that principal component analysis (PCA) is quite sensitive to outliers, with even a single corrupted data element carrying the potential of grossly biasing the recovered principal subspace. This is particularly true in many relevant applications that rely heavily on low-dimensional representations [8, 13, 27, 33, 22]. Mathematically, such outliers can be described by the measurement model $\mathbf{Y} = \mathbf{Z} + \mathbf{E}$, where $\mathbf{Y} \in \mathbb{R}^{n \times m}$ is an observed data matrix, $\mathbf{Z} = \mathbf{A}\mathbf{B}^\top$ is a low-rank component with principal subspace equal to span$[\mathbf{A}]$, and $\mathbf{E}$ is a matrix of unknown sparse corruptions with arbitrary amplitudes.

Ideally, we would like to remove the effects of $\mathbf{E}$, which would then allow regular PCA to be applied to $\mathbf{Z}$ for obtaining principal components devoid of unwanted bias. For this purpose, robust PCA (RPCA) algorithms have recently been motivated by the optimization problem

$$\min_{\mathbf{Z},\mathbf{E}} \ \max(n, m) \cdot \text{rank}[\mathbf{Z}] + \|\mathbf{E}\|_0 \ \text{ s.t. } \mathbf{Y} = \mathbf{Z} + \mathbf{E}, \tag{1}$$

where $\|\cdot\|_0$ denotes the $\ell_0$ matrix norm (meaning the number of nonzero matrix elements) and the $\max(n, m)$ multiplier ensures that both rank and sparsity terms scale between 0 and $nm$, reflecting a priori agnosticism about their relative contributions to $\mathbf{Y}$. The basic idea is that if $\{\mathbf{Z}^*, \mathbf{E}^*\}$ minimizes (1), then $\mathbf{Z}^*$ is likely to represent the original uncorrupted data.

As a point of reference, if we somehow knew *a priori* which elements of $\mathbf{E}$ were zero (i.e., no gross corruptions), then (1) could be effectively reduced to the much simpler matrix completion (MC) problem [5]

$$\min_{\mathbf{Z}} \text{rank}[\mathbf{Z}] \ \text{ s.t. } y_{ij} = z_{ij}, \ \forall (i,j) \in \Omega, \tag{2}$$

where $\Omega$ denotes the set of indices corresponding with zero-valued elements in $\mathbf{E}$. A major challenge with RPCA is that an accurate estimate of the support set $\Omega$ can be elusive.

Unfortunately, solving (1) is non-convex, discontinuous, and NP-hard in general. Therefore, the convex surrogate referred to as principal component pursuit (PCP)

$$\min_{\mathbf{Z},\mathbf{E}} \ \sqrt{\max(n,m)} \cdot \|\mathbf{Z}\|_* + \|\mathbf{E}\|_1 \ \ \text{s.t.} \ \mathbf{Y} = \mathbf{Z} + \mathbf{E} \tag{3}$$

is often adopted, where $\|\cdot\|_*$ denotes the nuclear norm and $\|\cdot\|_1$ is the $\ell_1$ matrix norm. These represent the tightest convex relaxations of the rank and $\ell_0$ norm functions respectively. Several theoretical results quantify technical conditions whereby the solutions of (1) and (3) are actually equivalent [4, 6]. However, these conditions are highly restrictive and do not provably hold in practical situations of interest such as face clustering [10], motion segmentation [10], high dynamic range imaging [22] or background subtraction [4]. Moreover, both the nuclear and $\ell_1$ norms are sensitive to data variances, often over-shrinking large singular values of $\mathbf{Z}$ or coefficients in $\mathbf{E}$ [11].

All of this motivates stronger approaches to approximating (1). In Section 2 we review existing alternatives, including both non-convex and probabilistic approaches; however, we argue that none of these can significantly outperform PCP in terms of principal subspace recovery in important, representative experimental settings devoid of prior knowledge (e.g., true signal distributions, outlier locations, rank, etc.). We then derive a new pseudo-Bayesian algorithm in Section 3 that has been tailored to conform with principled overarching design criteria. By 'pseudo', we mean an algorithm inspired by Bayesian modeling conventions, but with special modifications that deviate from the original probabilistic script for reasons related to estimation quality and computational efficiency. Next, Section 4 examines relevant theoretical properties, explicitly accounting for all approximations involved, while Section 5 provides empirical validations. Proofs and other technical details are deferred to [23]. Our high-level contributions can be summarized as follows:

- We derive a new pseudo-Bayesian RPCA algorithm with efficient ADMM subroutine.

- While provable recovery guarantees are absent for non-convex RPCA algorithms, we nonetheless quantify how our pseudo-Bayesian design choices lead to a desirable energy landscape. In particular, we show that although any outlier support pattern will represent an inescapable local minima of (1) (or a broad class of functions that mimic (1)), our proposal can simultaneously retain the correct global optimum while eradicating at least some of the suboptimal minima associated with incorrect outlier location estimates.

- We empirically demonstrate improved performance over state-of-the-art algorithms (including PCP) in terms of standard phase transition plots with a dramatically expanded success region. Quite surprisingly, our algorithm can even outperform convex matrix completion (MC) despite the fact that the latter is provided with *perfect knowledge of which entries are not corrupted*, suggesting that robust outlier support pattern estimation is indeed directly facilitated by our model.

## 2 Recent Work

The vast majority of algorithms for solving (1) either implicitly or explicitly attempt to solve a problem of the form

$$\min_{\mathbf{Z},\mathbf{E}} \ f_1(\mathbf{Z}) + \sum_{i,j} f_2(e_{ij}) \ \ \text{s.t.} \ \mathbf{Y} = \mathbf{Z} + \mathbf{E}, \tag{4}$$

where $f_1$ and $f_2$ are penalty functions that favor minimal rank and sparsity respectively. When $f_1$ is the nuclear norm (scaled appropriately) and $f_2(e) = |e|$, then (4) reduces to (3). Methods differ however by replacing $f_1$ and $f_2$ with non-convex alternatives, such as generalized Huber functions [7] or Schatten $\ell_p$ quasi-norms with $p < 1$ [18, 19]. When applied to the singular values of $\mathbf{Z}$ and elements of $\mathbf{E}$ respectively, these selections enact stronger enforcement of minimal rank and sparsity. If prior knowledge of the true rank of $\mathbf{Z}$ is available, a truncated nuclear norm approach (TNN-RPCA) has also been proposed [24]. Further divergences follow from the spectrum of optimization schemes applied to different objectives, such as the alternating directions method of multipliers (ADMM) algorithm [3] or iteratively reweighted least squares (IRLS) [18].

With all of these methods, we may consider relaxing the strict equality constraint to the regularized form

$$\min_{\mathbf{Z},\mathbf{E}} \ \tfrac{1}{\lambda}\|\mathbf{Y} - \mathbf{Z} - \mathbf{E}\|_{\mathcal{F}}^2 + f_1(\mathbf{Z}) + \sum_{i,j} f_2(e_{ij}), \tag{5}$$

where $\lambda > 0$ is a trade-off parameter. This has inspired a number of competing Bayesian formulations, which typically proceed as follows. Let

$$p(\mathbf{Y}|\mathbf{Z},\mathbf{E}) \ \propto \ \exp\left[-\tfrac{1}{2\lambda}\|\mathbf{Y} - \mathbf{Z} - \mathbf{E}\|_{\mathcal{F}}^2\right] \tag{6}$$

define a likelihood function, where $\lambda$ represents a non-negative variance parameter assumed to be known.[2] Hierarchical prior distributions are then assigned to $\mathbf{Z}$ and $\mathbf{E}$ to encourage minimal rank and strong sparsity, respectively. For the latter, the most common choice is the Gaussian scale-mixture (GSM) defined hierarchically by

$$p(\mathbf{E}|\mathbf{\Gamma}) = \prod_{i,j} p(e_{ij}|\gamma_{ij}), \quad p(e_{ij}|\gamma_{ij}) \propto \exp\left[-\frac{e_{ij}^2}{2\gamma_{ij}}\right], \quad \text{with hyper prior } p(\gamma_{ij}^{-1}) \propto \gamma_{ij}^{1-a} \exp[\frac{-b}{\gamma_{ij}}],$$
(7)

where $\mathbf{\Gamma}$ is a matrix of non-negative variances and $a, b \geq 0$ are fixed parameters. Note that when these values are small, the resulting distribution over each $e_{ij}$ (obtained by marginalizing over the respective $\gamma_{ij}$) is heavy-tailed with a sharp peak at zero, the defining characteristics of sparse priors.

For the prior on $\mathbf{Z}$, Bayesian methods have somewhat broader distinctions. In particular, a number of methods explicitly assume that $\mathbf{Z} = \mathbf{A}\mathbf{B}^\top$ and specify GSM priors on $\mathbf{A}$ and $\mathbf{B}$ [1, 9, 15, 30]. For example, variational Bayesian RPCA (VB-RPCA) [1] assumes $p(\mathbf{A}|\boldsymbol{\theta}) \propto \exp[-\text{tr}\left(\mathbf{A}\,\text{diag}[\boldsymbol{\theta}]^{-1}\mathbf{A}^\top\right)]$, where $\boldsymbol{\theta}$ is a non-negative variance vector. An equivalent prior is used for $p(\mathbf{B}|\boldsymbol{\theta})$ with a shared value of $\boldsymbol{\theta}$. This model also applies the prior $p(\boldsymbol{\theta}) = \prod_i p(\theta_i)$ with $p(\theta_i)$ defined for consistency with $p(\gamma_{ij}^{-1})$ in (7). Low rank solutions are favored via the same mechanism as described above for sparsity, but only the sparse variance prior is applied to *columns* of $\mathbf{A}$ and $\mathbf{B}$, effectively pruning them from the model if the associated $\theta_i$ is small. Given the above, the joint distribution is

$$p(\mathbf{Y}, \mathbf{A}, \mathbf{B}, \mathbf{E}, \mathbf{\Gamma}, \boldsymbol{\theta}) = p(\mathbf{Y}|\mathbf{A}, \mathbf{B}, \mathbf{E})p(\mathbf{E}|\mathbf{\Gamma})p(\mathbf{A}|\boldsymbol{\theta})p(\mathbf{B}|\boldsymbol{\theta})p(\mathbf{\Gamma})p(\boldsymbol{\theta}).$$
(8)

Full Bayesian inference with this is intractable, hence a common variational Bayesian (VB) mean-field approximation is applied [1, 2]. The basic idea is to obtain a tractable approximate factorial posterior distribution by solving

$$\min_{q(\mathbf{\Phi})} \mathcal{KL}\left[q(\mathbf{\Phi})||p(\mathbf{A}, \mathbf{B}, \mathbf{E}, \mathbf{\Gamma}, \boldsymbol{\theta}|\mathbf{Y})\right],$$
(9)

where $q(\mathbf{\Phi}) \triangleq q(\mathbf{A})q(\mathbf{B})q(\mathbf{E})q(\mathbf{\Gamma})q(\boldsymbol{\theta})$, each $q$ represents an arbitrary probability distribution, and $\mathcal{KL}[\cdot||\cdot]$ denotes the Kullback-Leibler divergence between two distributions. This can be accomplished via coordinate descent minimization over each respective $q$ distribution while holding the others fixed. Final estimates of $\mathbf{Z}$ and $\mathbf{E}$ are obtained by the means of $q(\mathbf{A})$, $q(\mathbf{B})$, and $q(\mathbf{E})$ upon convergence. A related hierarchical model is used in [9, 30], but MCMC sampling techniques are used for full Bayesian inference RPCA (FB-RPCA) at the expense of considerable computational complexity and multiple tuning parameters.

An alternative empirical Bayesian algorithm (EB-RPCA) is described in [31]. In addition to the likelihood function (6) and prior from (7), this method assumes a direct Gaussian prior on $\mathbf{Z}$ given by

$$p(\mathbf{Z}|\mathbf{\Psi}) \propto \exp\left[-\frac{1}{2}\text{tr}\left(\mathbf{Z}^\top\mathbf{\Psi}^{-1}\mathbf{Z}\right)\right],$$
(10)

where $\mathbf{\Psi}$ is a symmetric and positive definite matrix.[3] Inference is accomplished via an empirical Bayesian approach [20]. The basic idea is to marginalize out the unknown $\mathbf{Z}$ and $\mathbf{E}$ and solve

$$\max_{\mathbf{\Psi}, \mathbf{\Gamma}} \iint p(\mathbf{Y}|\mathbf{Z}, \mathbf{E})p(\mathbf{Z}|\mathbf{\Psi})p(\mathbf{E}|\mathbf{\Gamma})d\mathbf{Z}d\mathbf{E}$$
(11)

using an EM-like algorithm. Once we have an optimal $\{\mathbf{\Psi}^*, \mathbf{\Gamma}^*\}$, we then compute the posterior mean of $p(\mathbf{Z}, \mathbf{E}|\mathbf{Y}, \mathbf{\Psi}^*, \mathbf{\Gamma}^*)$ which is available in closed-form.

Finally, a recent class of methods has been derived around the concept of approximate message passing, AMP-RPCA [26], which applies Gaussian priors to the factors $\mathbf{A}$ and $\mathbf{B}$ and infers posterior estimates by loopy belief propagation [21]. In our experiments (see [23]) we found AMP-RPCA to be quite sensitive to data deviating from these distributions.

## 3  A New Pseudo-Bayesian Algorithm

As it turns out, it is quite difficult to derive a fully Bayesian model, or some tight variational/empirical approximation, that leads to an efficient algorithm capable of consistently outperforming the original convex PCP, at least in the absence of additional, exploitable prior knowledge. It is here that we adopt

a pseudo-Bayesian approach, by which we mean that a Bayesian-inspired cost function will be altered using manipulations that, although not consistent with any original Bayesian model, nonetheless produce desirable attributes relevant to blindly solving (1). In some sense however, we view this as a strength, because the final model analysis presented later in Section 4 does not rely on any presumed validity of the underlying prior assumptions, but rather on explicit properties of the objective that emerges, including all assumptions and approximation involved.

**Basic Model:** We begin with the same likelihood function from (6), noting that in the limit as $\lambda \to 0$ this will enforce the constraint set from (1). We also adopt the same prior on $\mathbf{E}$ given by (7) above and used in [1] and [31], but we need not assume any additional hyperprior on $\boldsymbol{\Gamma}$. In contrast, for the prior on $\mathbf{Z}$ our method diverges, and we define the Gaussian

$$p(\mathbf{Z}|\boldsymbol{\Psi}_r, \boldsymbol{\Psi}_c) \propto \exp\left[ -\tfrac{1}{2}\vec{z}^\top \left( \boldsymbol{\Psi}_r \otimes \mathbf{I} + \mathbf{I} \otimes \boldsymbol{\Psi}_c \right)^{-1} \vec{z} \right], \tag{12}$$

where $\vec{z} \triangleq \mathrm{vec}[\mathbf{Z}]$ is the column-wise vectorization of $\mathbf{Z}$, $\otimes$ denotes the Kronecker product, and $\boldsymbol{\Psi}_c \in \mathbb{R}^{n \times n}$ and $\boldsymbol{\Psi}_r \in \mathbb{R}^{m \times m}$ are positive semi-definite, symmetric matrices.[4] Here $\boldsymbol{\Psi}_c$ can be viewed as applying a column-wise covariance factor, and $\boldsymbol{\Psi}_r$ a row-wise one. Note that if $\boldsymbol{\Psi}_r = 0$, then this prior collapses to (10); however, by including $\boldsymbol{\Psi}_r$ we can retain symmetry in our model, or invariance to inference using either $\mathbf{Y}$ or $\mathbf{Y}^\top$. Related priors can also be used to improve the performance of affine rank minimization problems [34].

We apply the empirical Bayesian procedure from (11); the resulting convolution of Gaussians integral [2] can be computed in closed-form. After applying $-2\log[\cdot]$ transformation, this is equivalent to minimizing

$$\mathcal{L}(\boldsymbol{\Psi}_r, \boldsymbol{\Psi}_c, \boldsymbol{\Gamma}) = \vec{y}^\top \boldsymbol{\Sigma}_y^{-1} \vec{y} + \log|\boldsymbol{\Sigma}_y|, \qquad \text{where} \quad \boldsymbol{\Sigma}_y \triangleq \boldsymbol{\Psi}_r \otimes \mathbf{I} + \mathbf{I} \otimes \boldsymbol{\Psi}_c + \bar{\boldsymbol{\Gamma}} + \lambda \mathbf{I}, \tag{13}$$

and $\bar{\boldsymbol{\Gamma}} \triangleq \mathrm{diag}[\vec{\gamma}]$. Note that for even reasonably sized problems $\boldsymbol{\Sigma}_y \in \mathbb{R}^{nm \times nm}$ will be huge, and consequently we will require certain approximations to produce affordable update rules. Fortunately this can be accomplished while simultaneously retaining a principled objective function capable of outperforming existing methods.

**Pseudo-Bayesian Objective:** We first modify (13) to give

$$\mathcal{L}(\boldsymbol{\Psi}_r, \boldsymbol{\Psi}_c, \boldsymbol{\Gamma}) = \vec{y}^\top \boldsymbol{\Sigma}_y^{-1} \vec{y} + \sum_j \log\left| \boldsymbol{\Psi}_c + \tfrac{1}{2}\Gamma_{\cdot j} + \tfrac{\lambda}{2}\mathbf{I} \right| + \sum_i \log\left| \boldsymbol{\Psi}_r + \tfrac{1}{2}\Gamma_{i\cdot} + \tfrac{\lambda}{2}\mathbf{I} \right|, \tag{14}$$

where $\Gamma_{\cdot j} \triangleq \mathrm{diag}[\boldsymbol{\gamma}_{\cdot j}]$ and $\boldsymbol{\gamma}_{\cdot j}$ represents the $j$-th column of $\boldsymbol{\Gamma}$. Similarly we define $\Gamma_{i\cdot} \triangleq \mathrm{diag}[\boldsymbol{\gamma}_{i\cdot}]$ with $\boldsymbol{\gamma}_{i\cdot}$ the $i$-th row of $\boldsymbol{\Gamma}$. This new cost is nothing more than (13) but with the $\log|\cdot|$ term split in half producing a lower bound by Jensen's inequality; the Kronecker product can naturally be dissolved under these conditions. Additionally, (14) represents a departure from our original Bayesian model in that there is no longer any direct empirical Bayesian or VB formulation that would lead to (14). Note that although this modification cannot be justified on strictly probabilistic terms, we will see shortly that it nonetheless still represents a viable cost function in the abstract sense, and lends itself to increased computational efficiency. The latter is an immediate effect of the drastically reduced dimensionality of the matrices inside the determinant. Henceforth (14) will represent the cost function that we seek to minimize; relevant properties will be handled in Section 4. We emphasize that all subsequent analysis is based directly upon (14), and therefore already accounts for the approximation step in advancing from (13). This is unlike other Bayesian model justifications relying on the legitimacy of the original full model, and yet then adopt various approximations that may completely change the problem.

**Update Rules:** Common to many empirical Bayesian and VB approaches, our basic optimization strategy involves iteratively optimizing upper bounds on (14) in the spirit of majorization-minimization [12]. At a high level, our goal will be to apply bounds which separate $\boldsymbol{\Psi}_c$, $\boldsymbol{\Psi}_r$, and $\boldsymbol{\Gamma}$ into terms of the general form $\log|\mathbf{X}| + \mathrm{tr}[\mathbf{A}\mathbf{X}^{-1}]$, the reason being that this expression has a simple global minimum over $\mathbf{X}$ given by $\mathbf{X}=\mathbf{A}$. Therefore the strategy will be to update the bound (parameterized by some matrix $\mathbf{A}$), and then update the parameters of interest $\mathbf{X}$.

Using standard conjugate duality relationships and variational bounding techniques [14][Chapter 4], it follows after some linear algebra that

$$\vec{\boldsymbol{y}}^{\top}\boldsymbol{\Sigma}_y^{-1}\vec{\boldsymbol{y}} \quad \leq \quad \tfrac{1}{\lambda}\|\mathbf{Y} - \mathbf{Z} - \mathbf{E}\|_{\mathcal{F}}^2 + \sum_{i,j}\tfrac{e_{ij}^2}{\gamma_{ij}} + \vec{\boldsymbol{z}}^{\top}\left(\boldsymbol{\Psi}_r\otimes\mathbf{I} + \mathbf{I}\otimes\boldsymbol{\Psi}_c\right)^{-1}\vec{\boldsymbol{z}} \tag{15}$$

for all $\mathbf{Z}$ and $\mathbf{E}$. For fixed values of $\boldsymbol{\Psi}_r$, $\boldsymbol{\Psi}_c$, and $\boldsymbol{\Gamma}$ we optimize this quadratic bound to obtain revised estimates for $\mathbf{Z}$ and $\mathbf{E}$, noting that exact equality in (15) is possible via the closed-form solution

$$\vec{\boldsymbol{z}} = \left(\boldsymbol{\Psi}_r\otimes\mathbf{I} + \mathbf{I}\otimes\boldsymbol{\Psi}_c\right)\boldsymbol{\Sigma}_y^{-1}\vec{\boldsymbol{y}}, \qquad \vec{\boldsymbol{e}} = \bar{\boldsymbol{\Gamma}}\boldsymbol{\Sigma}_y^{-1}\vec{\boldsymbol{y}}. \tag{16}$$

In large practical problems, (16) may become expensive to compute directly because of the high dimensional inverse involved. However, we may still find the optimum efficiently by an ADMM procedure described in [23].

We can also further bound the righthand side of (15) using Jensen's inequality as

$$\vec{\boldsymbol{z}}^{\top}\left(\boldsymbol{\Psi}_r\otimes\mathbf{I} + \mathbf{I}\otimes\boldsymbol{\Psi}_c\right)^{-1}\vec{\boldsymbol{z}} \leq \operatorname{tr}\left[\mathbf{Z}^{\top}\mathbf{Z}\boldsymbol{\Psi}_r^{-1} + \mathbf{Z}\mathbf{Z}^{\top}\boldsymbol{\Psi}_c^{-1}\right]. \tag{17}$$

Along with (15) this implies that for fixed values of $\mathbf{Z}$ and $\mathbf{E}$ we can obtain an upper bound which only depends on $\boldsymbol{\Psi}_r$, $\boldsymbol{\Psi}_c$, and $\boldsymbol{\Gamma}$ in a decoupled or separable fashion.

For the $\log|\cdot|$ terms in (14), we also derive convenient upper bounds using determinant identities and a first-order approximation, the goal being to find a representation that plays well with the previous decoupled bound for optimization purposes. Again using conjugate duality relationships, we can form the bound

$$
\begin{aligned}
\log\left|\boldsymbol{\Psi}_c + \tfrac{1}{2}\boldsymbol{\Gamma}_{\cdot j} + \tfrac{\lambda}{2}\mathbf{I}\right| \quad &\equiv \quad \log|\boldsymbol{\Psi}_c| + \log|\boldsymbol{\Gamma}_{\cdot j}| + \log|W(\boldsymbol{\Psi}_c,\boldsymbol{\Gamma}_{\cdot j})| \\
&\leq \quad \log|\boldsymbol{\Psi}_c| + \log|\boldsymbol{\Gamma}_{\cdot j}| + \operatorname{tr}\left[(\nabla_{\boldsymbol{\Psi}_c^{-1}}^{j})^{\top}\boldsymbol{\Psi}_c^{-1}\right] + (\nabla_{\boldsymbol{\Gamma}_{\cdot j}^{-1}}^{c})^{\top}\boldsymbol{\gamma}_{\cdot j}^{-1} + C,
\end{aligned}
\tag{18}
$$

where the inverse $\boldsymbol{\gamma}_{\cdot j}^{-1}$ is understood to apply element-wise, and $W(\boldsymbol{\Psi}_c,\boldsymbol{\Gamma}_{\cdot j})$ is defined as

$$W(\boldsymbol{\Psi}_c,\boldsymbol{\Gamma}_{\cdot j}) \triangleq \frac{1}{2\lambda}\begin{bmatrix} 2\mathbf{I} & \sqrt{2}\mathbf{I} \\ \sqrt{2}\mathbf{I} & \mathbf{I} \end{bmatrix} + \begin{bmatrix} \boldsymbol{\Psi}_c^{-1} & \mathbf{0} \\ \mathbf{0} & \Gamma_{\cdot j}^{-1} \end{bmatrix}. \tag{19}$$

Additionally, $C$ is a standard constant, which accompanies the first-order approximation to guarantee that the upper bound is tangent to the underlying cost function; however, its exact value is irrelevant for optimization purposes. Finally, the requisite gradients are defined as

$$\nabla_{\Gamma_{\cdot j}^{-1}}^{c} \triangleq \frac{\partial W(\boldsymbol{\Psi}_c,\boldsymbol{\Gamma}_{\cdot j})}{\partial\Gamma_{\cdot j}^{-1}} = \operatorname{diag}[\boldsymbol{\Gamma}_{\cdot j} - \tfrac{1}{2}\boldsymbol{\Gamma}_{\cdot j}(\mathbf{S}_c^j)^{-1}\boldsymbol{\Gamma}_{\cdot j}], \quad \nabla_{\boldsymbol{\Psi}_c^{-1}}^{j} \triangleq \frac{\partial W(\boldsymbol{\Psi}_c,\boldsymbol{\Gamma}_{\cdot j})}{\partial\boldsymbol{\Psi}_c^{-1}} = \boldsymbol{\Psi}_c - \boldsymbol{\Psi}_c(\mathbf{S}_c^j)^{-1}\boldsymbol{\Psi}_c, \tag{20}$$

where $\mathbf{S}_c^j \triangleq \boldsymbol{\Psi}_c + \tfrac{1}{2}\boldsymbol{\Gamma}_{\cdot j} + \tfrac{\lambda}{2}\mathbf{I}$. Analogous bounds can be derived for the $\log\left|\boldsymbol{\Psi}_r + \tfrac{1}{2}\boldsymbol{\Gamma}_{i\cdot} + \tfrac{\lambda}{2}\mathbf{I}\right|$ terms in (14).

These bounds are principally useful because all $\boldsymbol{\Psi}_c$, $\boldsymbol{\Psi}_r$, $\boldsymbol{\Gamma}_{\cdot j}$, and $\boldsymbol{\Gamma}_{i\cdot}$ factors have been decoupled. Consequently, with $\mathbf{Z}$, $\mathbf{E}$, and all the relevant gradients fixed, we can separately combine $\boldsymbol{\Psi}_c$-, $\boldsymbol{\Psi}_r$-, and $\boldsymbol{\Gamma}$-dependent terms from the bounds and then optimize independently. For example, combining terms from (17) and (18) involving $\boldsymbol{\Psi}_c$ for all $j$, this requires solving

$$\min_{\boldsymbol{\Psi}_c}\; m\log|\boldsymbol{\Psi}_c| + \operatorname{tr}\left[\sum_j(\nabla_{\boldsymbol{\Psi}_c^{-1}}^{j})^{\top}\boldsymbol{\Psi}_c^{-1} + \mathbf{Z}\mathbf{Z}^{\top}\boldsymbol{\Psi}_c^{-1}\right]. \tag{21}$$

Analogous cost functions emerge for $\boldsymbol{\Psi}_r$ and $\boldsymbol{\Gamma}$. All three problems have closed-form optimal solutions given by

$$\boldsymbol{\Psi}_c = \tfrac{1}{m}\left[\sum_j\nabla_{\boldsymbol{\Psi}_c^{-1}}^{j\,\top} + \mathbf{Z}\mathbf{Z}^{\top}\right], \quad \boldsymbol{\Psi}_r = \tfrac{1}{n}\left[\sum_i\nabla_{\boldsymbol{\Psi}_r^{-1}}^{i\,\top} + \mathbf{Z}^{\top}\mathbf{Z}\right], \quad \vec{\boldsymbol{\gamma}} = \vec{\boldsymbol{z}}^2 + \vec{\boldsymbol{u}}_c + \vec{\boldsymbol{u}}_r, \tag{22}$$

where the squaring operator is applied element-wise to $\vec{\boldsymbol{z}}$, $\vec{\boldsymbol{u}}_c \triangleq [\nabla_{\Gamma_{\cdot 1}^{-1}}^{c};\ldots;\nabla_{\Gamma_{\cdot m}^{-1}}^{c}]$, and analogously for $\vec{\boldsymbol{u}}_r$. One interesting aspect of (22) is that it forces $\boldsymbol{\Psi}_c \succeq \tfrac{1}{m}\mathbf{Z}\mathbf{Z}^{\top}$ and $\boldsymbol{\Psi}_r \succeq \tfrac{1}{n}\mathbf{Z}^{\top}\mathbf{Z}$, thus maintaining a balancing symmetry and preventing one or the other from possibly converging towards zero. This is another desirable consequence of using the bound in (17). To finalize then, the proposed pipeline, which we henceforth refer to as pseudo-Bayesian RPCA (PB-RPCA), involves the steps shown under Algorithm 1 in [23]. These can be implemented in such a way that the complexity is linear in $\max(n,m)$ and cubic in $\min(n,m)$.

# 4   Analysis of the PB-RPCA Objective

On the surface it may appear that the PB-RPCA objective (14) represents a rather circuitous route to solving (1), with no obvious advantage over the convex PCP relaxation from (3), or any other approach for that matter. However quite surprisingly, we prove in [23] that by simply replacing the $\log|\cdot|$ matrix operators in (14) with $\text{tr}[\cdot]$, the resulting function collapses exactly to convex PCP. So what at first appear as distant cousins are actually quite closely related objectives. Of course our work is still in front of us to explain why $\log|\cdot|$, and therefore the PB-RPCA objective by association, might display any particular advantage. This leads us to considerations of *relative concavity*, *non-separability*, and *symmetry* as described below in turn.

**Relative Concavity:** Although both $\log|\cdot|$ and $\text{tr}[\cdot]$ are concave non-decreasing functions of the singular values of symmetric positive definite matrices, and hence favor both sparsity of $\boldsymbol{\Gamma}$ and minimal rank of $\boldsymbol{\Psi}_r$ or $\boldsymbol{\Psi}_c$, the former is far more strongly concave (in the sense of relative concavity described in [25]). In this respect we may expect that $\log|\cdot|$ is less likely to over-shrink large values [11]. Moreover, applying a concave non-decreasing penalty to elements of $\boldsymbol{\Gamma}$ favors a sparse estimate, which in turn transfers this sparsity directly to $\mathbf{E}$ by virtue of the left multiplication by $\bar{\boldsymbol{\Gamma}}$ in (16). Likewise for the singular values of $\boldsymbol{\Psi}_c$ and $\boldsymbol{\Psi}_r$.

**Non-Separability:** While potentially desirable, the relative concavity distinction described above is certainly not sufficient to motivate why PB-RPCA might represent an effective RPCA approach, especially given the breadth of non-convex alternatives already in the literature. However, a much stronger argument can be made by exposing a fundamental limitation of all RPCA methods (convex or otherwise) that rely on minimization of generic penalties in the separable or additive form of (4). For this purpose, let $\Omega$ denote a set of indices that correspond with zero-valued elements in $\mathbf{E}$, such that $\mathbf{E}_\Omega = 0$ while all other elements of $\mathbf{E}$ are arbitrary nonzeros (it can equally be viewed as the complement of the support of $\mathbf{E}$). In the case of MC, $\Omega$ would also represent the set of observed matrix elements. We then have the following:

**Proposition 1.** *To guarantee that (4) has the same global optimum as (1) for all $\mathbf{Y}$ where a unique solution exists, it follows that $f_1$ and $f_2$ must be non-convex and no feasible descent direction can ever remove an index from or decrease the cardinality of $\Omega$.*

In [31] it has been shown that, under similar conditions, the gradient in a feasible direction at any zero-valued element of $\mathbf{E}$ must be infinite to guarantee a matching global optimum, from which this result naturally follows. The ramifications of this proposition are profound if we ever wish to produce a version of RPCA that can mimic the desirable behavior of much simpler MC problems with known support, or at least radically improve upon PCP with unknown outlier support. In words, Proposition 1 implies that under the stated global-optimality preserving conditions, if any element of $\mathbf{E}$ converges to zero during optimization with an arbitrary descent algorithm, it will remain anchored at zero until the end. Consequently, if the algorithm prematurely errs in setting the wrong element to zero, meaning the wrong support pattern has been inferred at *any* time during an optimization trajectory, it is impossible to ever recover, a problem naturally side-stepped by MC where the support is effectively known. Therefore, the adoption of separable penalty functions can be quite constraining and they are unlikely to produce sufficiently reliable support recovery.

But how does this relate to PB-RPCA? Our algorithm maintains a decidedly *non-separable* penalty function on $\boldsymbol{\Psi}_c$, $\boldsymbol{\Psi}_r$, and $\boldsymbol{\Gamma}$, which directly transfers to an implicit, non-separable regularizer over $\mathbf{Z}$ and $\mathbf{E}$ when viewed through the dual-space framework from [32].[5] By this we mean a penalty $f(\mathbf{Z},\mathbf{E}) \neq f_1(\mathbf{Z}) + f_2(\mathbf{E})$ for any functions $f_1$ and $f_2$, and with $\mathbf{Z}$ fixed, we have $f(\mathbf{Z},\mathbf{E}) \neq \sum_{i,j} f_{ij}(e_{ij})$ for any set of functions $\{f_{ij}\}$.

We now examine the consequences. Let $\Omega$ now denote a set of indices that correspond with zero-valued elements in $\boldsymbol{\Gamma}$, which translates into an equivalent support set for $\mathbf{Z}$ via (16). This then leads to quantifiable benefits:

**Proposition 2.** *The following properties hold w.r.t. the PB-RPCA objective (assuming $n = m$ for simplicity):*

• *Assume that a unique global solution to (1) exists such that either $\text{rank}[\mathbf{Z}] + \max_j \|\boldsymbol{e}_{\cdot j}\|_0 < n$ or $\text{rank}[\mathbf{Z}] + \max_i \|\boldsymbol{e}_{i \cdot}\|_0 < n$. Additionally, let $\{\boldsymbol{\Psi}_c^*, \boldsymbol{\Psi}_r^*, \boldsymbol{\Gamma}^*\}$ denote a globally minimizing solution to (14) and $\{\mathbf{Z}^*, \mathbf{E}^*\}$ the corresponding values of $\mathbf{Z}$ and $\mathbf{E}$ computed using (16). Then in the limit $\lambda \to 0$, $\mathbf{Z}^*$ and $\mathbf{E}^*$ globally minimize (1).*

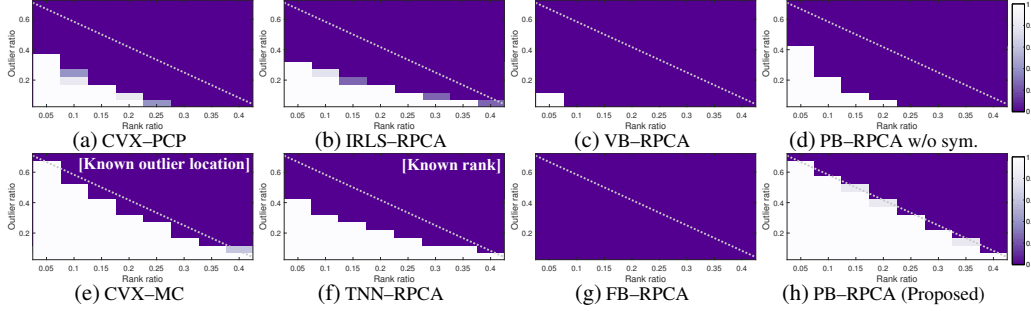

Figure 1: Phase transition over outlier (y-axis) and rank (x-axis) ratio variations. Here CVX-MC and TNN-RPCA maintain advantages of exactly known outlier support pattern and true rank respectively.

• *Assume that $\mathbf{Y}$ has no entries identically equal to zero.[6] Then for any arbitrary $\Omega$, there will always exist a range of $\mathbf{\Psi}_c$ and $\mathbf{\Psi}_r$ values such that for any $\mathbf{\Gamma}$ consistent with $\Omega$ we are not at a locally minimizing solution to (14), meaning there exists a feasible descent direction whereby elements of $\mathbf{\Gamma}$ can escape from zero.*

A couple important comments are worth stating regarding this result. First, the rank and row/column-sparsity requirements are extremely mild. In fact, *any* minimum of (1) will be such that $\text{rank}[\mathbf{Z}] + \max_j \|\boldsymbol{e}_{\cdot j}\|_0 \leq n$ and $\text{rank}[\mathbf{Z}] + \max_i \|\boldsymbol{e}_{i\cdot}\|_0 \leq m$, regardless of $\mathbf{Y}$. Secondly, unlike *any* separable penalty function (4) that retains the correct global optimal as (1), Proposition 2 implies that (14) need not be locally minimized by every possible support pattern for outlier locations. Consequently, premature convergence to suboptimal supports need not disrupt trajectories towards the global solution to the extent that (4) may be obstructed. Moreover, beyond algorithms that explicitly adopt separable penalties (the vast majority), some existing Bayesian approaches may implicitly default to (4). For example, as shown in [23], the mean-field factorizations adopted by VB-RPCA actually allow the underlying free energy objective to be expressible as (4) for some $f_1$ and $f_2$.

**Symmetry:** Without the introduction of symmetry via our pseudo-Bayesian proposal (meaning either $\mathbf{\Psi}_c$ or $\mathbf{\Psi}_r$ is forced to zero), then PB-RPCA collapses to something like EB-RPCA, which depends heavily on whether $\mathbf{Y}$ or $\mathbf{Y}^\top$ is provided as input and penalizes column- and row-spaces asymmetrically. In this regime it can be shown that the analogous requirement to replicate Proposition 2 becomes more stringent, namely we must assume the asymmetric condition $\text{rank}[\mathbf{Z}] + \max_j \|\boldsymbol{e}_{\cdot j}\|_0 < n$. Thus the symmetric cost of PB-RPCA of allows us to relax this column-wise restriction provided a row-wise alternative holds (and vice versa), allowing the PB-RPCA objective (14) to match the global optimum of our original problem from (1) under broader conditions.

In closing this section, we reiterate that all of our analysis and conclusions are based on (14), *after* the stated approximations. Therefore we need not rely on the plausibility of the original Bayesian starting point from Section 3 nor the tightness of subsequent approximations for justification; rather (14) can be viewed as a principled stand-alone objective for RPCA regardless of its origins. Moreover, it represents the first approach satisfying the relative concavity, non-separability, and symmetry properties described above, which can loosely be viewed as necessary, but not sufficient design criteria for an optimal RPCA objective.

## 5 Experiments

To examine significant factors that influence the ability to solve (1), we first evaluate the relative performance of PB-RPCA estimating random simulated subspaces from corrupted measurements, the standard benchmark. Later we present subspace clustering results for motion segmentation as a practical application. Additional experiments and a photometric stereo example are provided in [23].

**Phase Transition Graphs:** We compare our method against existing RPCA methods: PCP [16], TNN [24], IRLS [18], VB [1], and FB [9]. We also include results using PB-RPCA but with symmetry removed (which then defaults to something like EB-RPCA), allowing us to isolate the importance of this factor, called "PB-RPCA w/o sym.". For competing algorithms, we set parameters based on the values suggested by original authors with the exception of IRLS. Detailed settings and parameters can be found in [23].

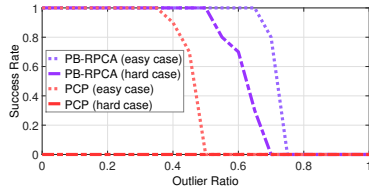

Figure 2: Hard case comparison.

| $\rho$ | SSC | Robust SSC | PCP+SSC | PB+SSC (Ours) |
|---|---|---|---|---|
| | Without sub-sampling (large number of measurements) | | | |
| 0.1 | 19.0 / 14.9 | 5.3 / 0.3 | 3.0 / 0.0 | 2.4 / 0.0 |
| 0.2 | 28.2 / 28.3 | 6.4 / 0.4 | 3.0 / 0.0 | 2.4 / 0.0 |
| 0.3 | 33.2 / 34.7 | 7.2 / 0.5 | 3.6 / 0.2 | 2.8 / 0.0 |
| 0.4 | 36.5 / 39.0 | 8.5 / 0.6 | 4.7 / 0.2 | 3.1 / 0.0 |
| | With sub-sampling (small number of measurements) | | | |
| 0.1 | 19.5 / 17.2 | 4.0 / 0.0 | 2.9 / 0.0 | 2.8 / 0.0 |
| 0.2 | 33.0 / 33.3 | 5.3 / 0.0 | 3.7 / 0.0 | 3.6 / 0.0 |
| 0.3 | 39.3 / 41.1 | 5.7 / 1.7 | 5.0 / 0.7 | 3.9 / 0.0 |
| 0.4 | 42.2 / 43.5 | 6.4 / 2.1 | 9.8 / 5.1 | 3.7 / 0.0 |

*Values are percentage with (mean / median).

Figure 3: Motion segmentation errors on Hopkins155.

We construct phase transition plots as in [4, 9] that evaluate the recovery success of every pairing of outlier ratio and rank using data $\mathbf{Y}=\mathbf{Z}_{GT}+\mathbf{E}_{GT}$, where $\mathbf{Y}\in\mathbb{R}^{m\times n}$ and $m=n=200$. The ground truth outlier matrix $\mathbf{E}_{GT}$ is generated by selecting non-zero entries uniformly with probability $\rho\in[0,1]$, and its magnitudes are sampled iid from the uniform distribution $U[-20, 20]$. We generate the ground truth low-rank matrix by $\mathbf{Z}_{GT}=\mathbf{A}\mathbf{B}^\top$, where $\mathbf{A}\in\mathbb{R}^{n\times r}$ and $\mathbf{B}\in\mathbb{R}^{m\times r}$ are drawn from iid $\mathcal{N}(0,1)$.

Figure 1 shows comparisons among competing methods, as well as the convex nuclear norm based matrix completion (CVX-MC) [5], the latter representing a *far easier* estimation task given that missing entry locations (analogous to corruptions) occur in known locations. The color of each cell encodes the percentage of success trials (out of 10 total) whereby the normalized root-mean-squared error (NRMSE, $\frac{\|\hat{\mathbf{Z}}-\mathbf{Z}_{GT}\|_\mathcal{F}}{\|\mathbf{Z}_{GT}\|_\mathcal{F}}$) recovering $\mathbf{Z}_{GT}$ is less than $0.001$ to classify success following [4, 9].

Notably PB-RPCA displays a much broader recoverability region. This improvement is even maintained over TNN-RPCA and MC which require prior knowledge such as the true rank and exact outlier locations respectively. These forms of prior knowledge offer a substantial advantage, although in practical situations are usually unavailable. PB-RPCA also outperforms PB-RPCA w/o sym. (its closest relative) by a wide margin, suggesting that the symmetry plays an important role. The poor performance of FB-RPCA is explained in [23].

**Hard Case Comparison:** Recovery of Gaussian iid low-rank components (the typical benchmark recovery problem in the literature) is somewhat ideal for existing algorithms like PCP because the singular vectors of $\mathbf{Z}_{GT}$ will not resemble unit vectors that could be mistaken for sparse components. However, a simple test reveals just how brittle PCP is to deviations from the theoretically optimal regime. We generate a rank one $\mathbf{Z}_{GT} = \sigma\mathbf{a}^3(\mathbf{b}^3)^\top$, where the cube operation is applied element-wise, $\mathbf{a}$ and $\mathbf{b}$ are vectors drawn iid from a unit sphere, and $\sigma$ scales $\mathbf{Z}_{GT}$ to unit variance. $\mathbf{E}_{GT}$ has nonzero elements drawn iid from $U[-1, 1]$. Figure 2 shows the recovery results as the outlier ratio is increased. The *hard case* refers to the data just described, while the *easy case* follows the model used to make the phase transition plots. While PB-RPCA is quite stable, PCP completely fails for the hard data.

**Outlier Removal for Motion Segmentation:** Under an affine camera model, the stacked matrix consisting of feature point trajectories of $k$ rigidly moving objects forms a union of $k$ affine subspaces of at most rank $4k$ [29]. But in practice, mismatches often occur due to occlusions or tracking algorithm limitations, and these introduce significant outliers into the feature motions such that the corresponding trajectory matrix may be at or near full rank. We adopt an experimental paradigm from [17] designed to test motion segmentation estimation in the presence of outliers. To mimic mismatches while retaining access to ground-truth, we randomly corrupt the entries of the trajectory matrix formed from Hopkins155 data [28]. Specifically, following [17] we add noise drawn from $\mathcal{N}(0, 0.1\kappa)$ to randomly sampled points with outlier ratio $\rho\in[0, 1]$, where $\kappa$ is the maximum absolute value of the data. We may then attempt to recover a clean version from the corrupted measurements using RPCA as a preprocessing step; motion segmentation can then be applied using standard subspace clustering [29]. We use SSC and robust SSC algorithms [10] as baselines, and compare with RPCA preprocessing computed via PCP (as suggested in [10]) and PB-RPCA followed by SSC. Additionally, we sub-sampled the trajectory matrix to increase problem difficulty by fewer samples. Segmentation accuracy is reported in Fig. 3, where we observe that PB shows the best performance across different outlier ratios, and the performance gap widens when the measurements are scarce.

## 6  Conclusion

Since the introduction of convex RPCA algorithms, there has not been a significant algorithmic break-through in terms of dramatically enhancing the regime where success is possible, at least in the absence of any prior information (beyond the generic low-rank and sparsity assumptions). The likely explanation is that essentially all of these approaches solve either a problem in the form of (4), an asymmetric problem in the form of (11), or else require strong priori knowledge. We provide a novel integration of three important design criteria, concavity, non-separability, and symmetry, that leads to state-of-the-art results by a wide margin *without tuning parameters or prior knowledge*.

## Footnotes

*This work was done while the first author was an intern at Microsoft Research, Beijing. The first and third authors were supported by the NRF of Korea grant funded by the Korea government, MSIP (No. 2010-0028680). The second author was partly supported by JSPS KAKENHI Grant Number JP16H01732.

[2]Actually many methods attempt to learn this parameter from data, but we avoid this consideration for simplicity. As well, for subtle reasons such learning is sometimes not even identifiable in the strict statistical sense.

[3]Note that in [31] this method is motivated from an entirely different variational perspective anchored in convex analysis; however, the cost function that ultimately emerges is equivalent to what follows with these priors.

[4]Technically the Kronecker sum $\boldsymbol{\Psi}_r \otimes \mathbf{I} + \mathbf{I} \otimes \boldsymbol{\Psi}_c$ must be positive definite for the inverse in (12) to be defined. However, we can accommodate the semi-definite case using the following convention. Without loss of generality assume that $\boldsymbol{\Psi}_r \otimes \mathbf{I} + \mathbf{I} \otimes \boldsymbol{\Psi}_c = \mathbf{R}\mathbf{R}^\top$ for some matrix $\mathbf{R}$. We then qualify that $p(\mathbf{Z}|\boldsymbol{\Psi}_r, \boldsymbol{\Psi}_c) = 0$ if $\vec{z} \notin \mathrm{span}[\mathbf{R}]$, and $p(\mathbf{Z}|\boldsymbol{\Psi}_r, \boldsymbol{\Psi}_c) \propto \exp[-\tfrac{1}{2}\vec{z}^\top (\mathbf{R}^\top)^\dagger \mathbf{R}^\dagger \vec{z}]$ otherwise.

[5]Even though this penalty function is not available in closed-form, non-separability is nonetheless enforced via the linkage between $\boldsymbol{\Psi}_c$, $\boldsymbol{\Psi}_r$, and $\boldsymbol{\Gamma}$ in the $\log|\cdot|$ operator.

[6]This assumption can be relaxed with some additional effort but we avoid such considerations here for clarity of presentation.

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
