[Reviews · NeurIPS 2016]

Reviewer 1

Summary

The paper presents a variational algorithm for robust Bayesian PCA.

Qualitative Assessment

Update after author feedback: I thank the authors for the clarification of the purpose of the paper. Including that in the text might make it easier for all your readers to appreciate the paper. --------------------------------------- Technical quality: The method seems reasonable. The experiments are weak: all the experiments appear at least partly synthetic in terms of added noise. Novelty: The method is an interesting novel combination of existing ideas. The main novelty appears to be a new bound on a joint column-wise and row-wise prior. Impact: On paper, the method looks very good and one would expect it to attract a lot of users. I am however underwhelmed by the experiments: the authors have do not seem to have been able to find a real practical killer application for their method which raises some doubt of how useful it will be in practice. Clarity: The paper is difficult to read because there appears to be too much material crammed in. There are many comments that could be expanded or supported by relevant references (e.g. lines 34, 82-83, Eq. (7) and its relation to ARD, line 135, proof of Proposition 1, ...). I feel the submission might work better as a journal paper than a brief conference paper. I am not sure why the authors have chosen to call their method pseudo-Bayesian, which seems like a very uninformative name. The method seems like a good examples of a variational algorithm in the spirit of Jordan et al., i.e. not of the KL divergence minimising type. At least I would find the term "variational" much more informative as a characterisation. The experiments are described very briefly and many important parameters and their values are not described, especially how r is chosen. The metrics used in all the experiments should be presented and the parameters defining success justified. The sensitivity of the results to any arbitrary parameters should be analysed. There is no discussion on weaknesses and limitations of the method.

Confidence in this Review

2-Confident (read it all; understood it all reasonably well)


Reviewer 2

Summary

The paper describes a objective function and optimization approach for robust PCA, where the goal is to find a low-rank subspace explaining the data in the presence of corrupted data values. The paper introduces the problem, reviews related methods, presents a new algorithm beginning from a Bayesian model but modifying the objective for better computational tractability, argues for the desirable properties of this objective, and presents experiments in simulated data and motion segmentation data with simulated corruption. Compared to select previous approaches, the results are impressively better with regard to shown success rate in recovering the low-rank subspace.

Qualitative Assessment

The writing suffers from the page limit and seems condensed (to me) and I have to admit not being able to follow all the steps in the development of the computational algorithm, and the explicit argumentation where the previous approaches fail is not always entirely clear. (Unfortunately, my review may be thus of limited value.) Comments: 1) How does the prior on Z (eqn 12) compare to matrix normal? Why (12) instead of matrix normal (that is, why Psi_r (x) I + I (x) Psi_c instead of Psi_r (x) Psi_c)? 2) Does the unmodified objective (13) realize the design criteria (I don't see why it wouldn't...)? Would this, if it was computationally tractable, more more desirable objective as it conforms to clear model assumptions that could be revised if there is prior information or if the method does not work well for some data? 3) How fast and stable is the computation in practice? Can this scale to large problems?

Confidence in this Review

1-Less confident (might not have understood significant parts)


Reviewer 3

Summary

This paper derives a robust PCA approach, inspired by a empirically simplified Bayesian model. Contrary to most of the probabilistic/Bayesian counterparts of robust PCA which generally resort to approximations during the inference scheme, the authors introduce an empirical (so-called pseudo-Bayesian) objective function, derived as a simpler surrogate of the posterior. This objective function is finally minimized by an iterative optimization strategy inspired by majorization-minimization schemes. Interestingly, a in-depth analysis of the surrogate objective function is conducted with respect to the initial Bayesian model, which tends to properly justify the interest of the proposed approach. This interest is illustrated by extensive experiments which compare several state-of-the-art algorithms, while putting a significant effort to understand why the proposed approach significantly outperforms the others. This reviewer enjoys reading this interesting papers and would like to congratulate the authors for their honesty while deriving the proposed objective function and theirs efforts to properly analyze it.

Qualitative Assessment

1. As said earlier, the reviewer particularly appreciated the way the authors introduce this pseudo-Bayesian objective function. More importantly, Section 4 provides a convincing lighting on this criterion with accurate and relevant arguments. 2. The results reported in the experimental section are impressive. The reviewer also appreciated the efforts dedicated to understand why the FB-RPCA fails to provide reliable results, in spite of its popularity. Given the high quality of this paper, the reviewer has not any additional comment/advice to give to the authors whose paper introduces an interesting alternative of PCP.

Confidence in this Review

2-Confident (read it all; understood it all reasonably well)


Reviewer 4

Summary

This paper introduces a new Bayesian framework for robust PCA. The optimization formulation proposed is inspired prior fully Bayesian framework but has much simpler structure. The analysis shows that by replacing a certain term in their objective function, the method is actually equivalent to convex PCP. Then several comments regarding why this trick is better are provided. The experiments show better empirical performance of the proposed method over prior arts.

Qualitative Assessment

The formulation for robust PCA given in this paper is shown to have better empirical performance, which I think would be one of the key contributions of this paper. The new model is inspired by existing Bayesian framework, but simplifies some probabilistic dependencies. I doubt that such modifications might contain little novelty, and the algorithmic contribution of this paper is incremental. But I am not an expert in Bayesian models, so not sure about this point. The analysis given in this paper doesn't convince me why the proposed method is better than existing Bayesian RPCA and also PCP. What is the motivation to show that the proposed method is equivalent to convex PCP under certain situations? If so, is it possible to obtain solid theory like convex PCP? Overall, this paper could be a good empirical work considering their empirical improvement, but may not have solid theory foundation as claimed.

Confidence in this Review

1-Less confident (might not have understood significant parts)


Reviewer 5

Summary

This paper is presenting a a method for solving robust PCA problem inspired by Bayesian formulation. Solving this formulation of the problem improves the region of inputs where successful recovery of the sparse and low-rank matrices is possible.

Qualitative Assessment

The results definitely look promising. The pseudo Bayesian formulation has led to a difficult optimization problem which requires a lot of math and therefore having a lot of equations is inevitable. Nevertheless, it is sometimes difficult to follow the equations and particularly the update rules. For those not working specifically in this field, it is not easy to understand (e.g. it might not be obvious to everyone how equation (12) is encouraging a low rank matrix, etc.). I suggest adding a brief summary of what you are doing without getting involved too much in the math. The results and the fact that such a difficult optimization problem can be solved efficiently is very interesting though.

Confidence in this Review

2-Confident (read it all; understood it all reasonably well)


Reviewer 6

Summary

This paper proposes a Bayes-inspired (though not Bayesian) approach to RPCA. They offer a tractable algorithm that can be used to perform this RPCA. While their program is nonconvex, they spend considerable effort motivating their decisions and highlighting the desirable properties introduced by them. They offer simulations that demonstrate improved performance and an expanded success regime over existing RPCA programs.

Qualitative Assessment

Similar to the hard case where PB-RPCA is better than traditional RPCA, could you provide any cases that traditional RPCA might outperform the proposed PB-RPCA if there is any. 'update'-------------------- I think the authors argued that PB-RPCA is always better than traditional RPCA is not solid. I don't agree that initialization using traditional RPCA (a convex program) and then perform PB-RCPA will definitely yields better results than traditional RPCA, since the deep truth is that they just use different objective. So for some particular prior traditional RPCA is better than PB-RPCA. The paper would not be complete without point out this limitation. Also the paper seems to need some practical application to justify their PB-RPCA, even though they provided some synthetic experiments to justify the superior performance of PB-RPCA (probably under some favorable prior). Strength: The objective is new and might give some researchers new ideas in practice when solving RPCA problem. Then their justification would also add some knowledge to this line of work.

Confidence in this Review

1-Less confident (might not have understood significant parts)